# How the Built Environment Moderates Gender Gap in Active Commuting to Schools

**DOI:** 10.3390/ijerph20021131

**Published:** 2023-01-09

**Authors:** Masoud Javadpoor, Ali Soltani, Leila Fatehnia, Negin Soltani

**Affiliations:** 1School of Art and Architecture, Shiraz University, Shiraz 71946-84471, Iran; 2UniSA Business, University of South Australia, Adelaide, SA 5001, Australia; 3School of Civil Engineering, Shiraz University, Shiraz 71557-13876, Iran

**Keywords:** gender, school travel, built environment, walkability, Iran

## Abstract

This study investigates the influences of built environmental (BE) factors, network design, and sociodemographic factors on active school travel (AST). Although numerous studies have explored these relationships, this study is trying to assess this issue with a focus on gender differences. Data from a cross-sectional sample of children from first to sixth grades from 16 public primary schools exclusive for girls and boys (*N* = 1260) in Shiraz collected in November 2019 was used. The analysis of the data revealed that, on average, boys are more willing to walk than girls, but that the boys’ tendency to walk is less elastic with respect to distance. Moreover, it is shown that street connectivity for all distance thresholds has a positive relationship with walking level, but the street network choice parameter decreases the chance of walking within an 800 metre threshold. It is demonstrated the need to take gender differences into account in local planning policies to promote AST in a developing country context.

## 1. Introduction

Childhood obesity has become a substantial and problematic epidemic in the twenty-first century [1], affecting children’s growth and hastening severe cardiovascular and metabolic risk factors and psychological issues [1,2]. The frequency of childhood overweight and obesity has risen alarmingly in most developing nations [1]. Iran is one of these nations, with 7.9% and 5.6% of school-aged children being overweight or obese, respectively [1]. Moreover, childhood obesity is followed into adulthood, and its early beginning might increase mortality risk later in life [1]. As a result, physical exercise in childhood is critical for promoting a healthy lifestyle for life and lowering the prevalence of obesity and related chronic disorders [3]. According to the World Health Organization (WHO), children should engage in at least 60 min of moderate to intense physical exercise daily [4]. Active school travel (AST) is an option that may be used regularly throughout the week [2], which can be an useful chance to ensure that children perform adequate physical activity during the day [5]. Non-motorized transport, such as walking and cycling, and other active modes, such as non-electric scooters, are examples of AST [6]. AST promotion has been proven in studies to enhance children’s physical health [7], mental health [7], anxiety relief [8], increase positive emotions such as happiness, excitement, or relaxation [7,9], reduce obesity [7,10], and prevention of diabetes [10]. Participating in AST can also enhance children’s social interactions and interpersonal connections [11]. However, in recent years, we have seen a decline in AST in several middle- and high-income countries worldwide [6]. This issue has the potential to have a substantial impact on the health of children and adolescents in the future.

Numerous studies have shown that men and women travel differently [12]. For instance, the literature indicates that women are more sensitive to traffic dangers while using active transportation than males [13]. In addition, societal and cultural barriers and the fear of sexual harassment by males discourage women from engaging in active travel. As gender inequality differs among cultures, it is probable that the gender gap in travel patterns, particularly active travel, varies substantially globally [14]. AST is no exception to the governing rule.

While there have been a lot of studies conducted on AST, there are several fundamental constraints. The influence of physical environment variables on children’s travel pattern is a fundamental constraint of the absence of a good knowledge of the factors determining their AST, stressing gender disparities. Although a small number of research studies [15] have usually analyzed AST with an emphasis on children’s gender, little attention has been devoted to it. On the other hand, one of the primary assumptions underlying this study’s concentration on gender categories is that schools in Iran are gender segregated, with boys and girls attending different schools. Furthermore, one of the primary causes for gender variations in travel pattern in Iran is cultural differences. Although there are no limits on women’s physical activities and social contacts in Iran, Iranian families may be more conservative about their daughters and do not allow them to go to and from school as freely as boys [15,16]. As a result, the factors influencing the travel of boys and girls to school may differ. The influence of physical environment elements on children’s AST is another significant weakness in earlier active travel studies. Many studies have been conducted to assess the impact of the physical environment on children’s active travel, but little emphasis has been devoted to how they differ by gender. According to the findings, space syntax indicators are useful for assessing how urban street layout connects to mobility potentials [6]. However, it received little attention in the associated literature. In order to address the constraints of prior research, indicators of space syntax were employed in this study, along with other environmental variables, to seek a better knowledge of the relationship between urban design and space syntax. To fill the gaps, this study was conducted to analyze the physical environment variables impacting the active transit of children to school, with a specific focus on gender disparities. For achieving this goal, a range of analytical and statistical methods are applied using different software tools as depicted in the research process diagram (Figure 1).

## 2. Literature Review

Personal traits and, more significantly, socioeconomic characteristics, particularly those of parents, have been proven in studies to impact children’s choice of AST. Previous research has found that personal variables such as age [17,18] and gender [18,19] have a role in AST choice. Some studies, for example, claim that the likelihood of AST increases until the age of 10 and subsequently decreases [17,20]. Furthermore, it has been suggested that boys tend to travel more actively than girls [19,21,22]

The characteristics and personal attitudes of parents and families, for example, parents’ pattern in traveling to work or non-work travel, and their understanding and perspective regarding street safety and environmental security are among the interpersonal factors that influence children’s active travel [6,23]. However, investigations have had conflicting results. For example, some studies have indicated that children are impacted by their parents’ behavior patterns, including the way they travel, owing to the role their parents set. It was demonstrated that parents who walk more, their children are more likely to walk to school [24]. Nonetheless, no significant association was discovered between parents’ travel patterns and children’s active travel in another study [25].

Socioeconomic status influences a child’s AST choice [26]. Many studies, for example, have found that the features and status of parents’ work, income, car ownership, number of siblings in the family, and number of persons with driving license [15,27] influence AST levels. The evidence for this remains indeterminate. Some studies have found that ownership of a private car by a family lowers the quantity of AST in children [28]. In addition, some research has found a negative connection between family income and AST [17,26]. According to previous studies, having one or more older siblings in the household increases children’s independent movement and active travel [26]. The presence of an older sibling may accompany the child, providing protection and peace of mind for the parents as the children go to school [29]. However, several investigations have shown no significant association [15]. Another aspect influencing children’s school travel is their parents’ work type and pattern. The presence of an adult at home with flexible working hours enhances children’s active travel, according to previous research [30]. 

A substantial amount of prior research has indicated that the physical environment is a key component in shaping student travel patterns [6]. This suggests that the physical qualities of the urban environment influence children’s active travel [15,21]. However, the results are still varied, and its impact is not clearly characterized, particularly in terms of gender. Some research has established that the distance from the school is the most important factor in influencing the choice of travel pattern [31]. Thus, children who live further away from the school travel less actively to school [31]. However, there is minimal consensus in past studies on the ideal distance [31]. According to Iranian research, children’s school travel threshold is 10 min [32], roughly equal to 700 m if children’s average speed is 4.3 km/h [33]. A study in Spain also found that the average walking distance is 875 m for younger children and 1350 m for teenagers [34]. Another study found that a 1.6 km walk to school was a good starting point for promoting active travel among children in the United States [35]. Furthermore, the distance of physical qualities around the student’s house plays an important part in children’s AST [6]; hence the association of physical characteristics at different distances from home with AST must be considered.

An increasing body of research indicates that land use features and roadway network design can impact children walking to school [36]. For example, several studies have demonstrated that children who live in high-density neighborhoods walk less to school [6]. It has also been demonstrated that the land use mix enhances walking to school; this can be attributed to the land use mix boosting the attractiveness of the area for pedestrians [37]. However, there are still inconsistent findings regarding land use mix, with some reporting a positive association between land use mix and active travel [27,37], while others indicate a negative relationship [38].

The positive association of urban green space with physical activity and active mobility has been highlighted among built environmental elements [39]. According to several research studies, the Normalized Difference Vegetation Index (NDVI) as an objective measure of urban green space is associated with the likelihood of children engaging in physical activity [40]. Another research study found that increasing urban greenness (NDVI) in an 800-m radius surrounding a school enhances the chance of children engaging in active travel [39]. Some studies discovered a positive association between the desire for AST and the availability of trees and street parks [41], while others found a negative relationship between open and green areas and active travel [42]. Meanwhile, some studies did not show a significant relationship between NDVI and AST [43]. However, this relationship is still unclear, and the results are contradictory [44,45].

Growing research over the last decade has stressed the significance and influence of roadway network design on school travel [6]. Some studies [6,36] employed various measures to assess street network design. According to previous research studies, the high density of street crossings [22] and the small size of urban blocks [42] boost the likelihood of walking to school. Previous studies, however, have not adequately studied the link between street design and AST. The space syntax technique depicts the structure of the street network as described by the connectedness hierarchy as assessed by changes in direction, which is significant in children’s pedestrian travel and adult cycling [28,46]. The space syntax is an appropriate tool for critically examining street design and revealing crucial linkages with walking [6]. In fact, areas with strong spatial connectedness and permeability are more likely to attract pedestrians than other spaces, but they cannot quickly reach locations with poor connectivity (requiring a significant number of turns). Therefore pedestrians may be less drawn to such spaces [47]. However, the conclusions of how the indicators of space syntax impact active travel remain equivocal. According to the findings, constructing a well-connected street network allows for more direct and pleasurable active transport [36]. Furthermore, several research studies found that greater roadway network connection is strongly related to increased rates of walking to school [28]. Another study found that a roadway network with limited access lowers the likelihood of each individual walking or cycling instead of driving [48].

## 3. Materials and Methods

### 3.1. Area of Study

The study’s goal was to look at the influence of individual, economic, social, and built environment (BE) variables on the travel pattern of primary school students in Shiraz, with a focus on gender. Shiraz is Iran’s sixth most populated city, with a population of 1,565,572, according to the 2015 census [49]. This city has a total size of 240 square kilometers and a population density of 8240 persons per square kilometer [50]. Shiraz city is split administratively into 11 urban regions, each with its own municipal government. Shiraz’s physical development pattern is semi-linear [51]. Shiraz has seen an increase in motorized travel due to the significant increase in car ownership over the last decade, as well as the availability of cheap fuel (the average price of gasoline in the world is $1.06, while in Iran, it is $0.06), and the low quality of the infrastructure of the public transportation fleet. According to a Shiraz Household Travel survey, 50% of travel is by private car, 28% by taxi, 12% by metro and bus, and 10% by other modes (walking not included) [51]. In terms of morphology, the Shiraz metropolis may be split into three zones: internal, middle, and exterior [52]. Each zone has a distinct demographic and urban layout, which influences travel patterns [53]. Choosing schools in locations with varying urban form structures can provide a more complete investigation of the BE’s influence on inhabitants’ travel patterns [54].

Shiraz’s inner city has an organic texture dominated by administrative/governmental buildings, with a considerable number of residential units developed in the previous two decades. This area also contains substantial portions of traditional and historic communities, historical sites, and tourism destinations. The semi-organic and semi-checkered texture of the middle section contains heritage gardens in the west, as well as some wasteland and freshly refurbished structures [55]. Although there are sub-centers in this region, administrative and commercial concerns are mostly handled by Shiraz CBD. Shiraz’s outer area has a new and regular texture that has generally been developed in the last decade; some new communities are planned or under construction, and some are independent rural and peripheral settlements that have been dissolved into the body of the metropolis due to the city’s rapid growth [52,53]. Schools were chosen from all three zones to represent the whole metropolitan area of Shiraz (Figure 2). 

### 3.2. Sampling

This study’s sample and survey design were carried out grade by grade in November 2019 among students in the first to sixth grades (6 to 12 years old) in public primary schools for girls and boys. Initially, five municipality districts were chosen from Shiraz—municipality districts were selected from zones with different morphology- which varied in terms of inhabitants’ economic level, land use type, and physical characteristics, particularly in the structure of the street network (the influence of these elements on children’s travel has been explored in earlier research). Among these regions, 16 public primary schools for girls and boys were chosen at random (an attempt was made in each region to identify schools for girls and boys in settings with similar features). In the selected schools, one class was selected from each educational level, and then a questionnaire was presented to the students using random numbers. In 16 public schools for boys and girls, 2380 questionnaires were distributed. The 1490 questionnaires returned with a 63% response rate. A questionnaire, signed consent form, and a map (to determine the precise location of the house) were sent/collected from the parents for the students to complete. The Shiraz University Research Directory approved the study protocol, and the General Education Department of Shiraz and school principals issued the necessary approvals to visit the schools. Table 1 displays the contribution of active travel by gender and zone of students participating in the research study.

The students’ age, and home address were all requested in the first section of the questionnaire. In the second section of the questionnaire, parents were asked six questions, including their employment status (inflexible/flexible), the number of student children (7 to 18 years old) in the household (one, two, three, and more), household income (Less than $595, $595 to $1190, and More than $1190), the number of cars owned by the household (zero, one, two, three, or more), the number of people with a driving license (zero, one, two, three, or more), the student’s traveling pattern was asked in the third section of the questionnaire, which travel pattern do you normally use for traveling from home to school? Furthermore, parents pick between active (walking/cycling) and passive travel for their children (using public transportation, school service, private car, etc.) (Table 2). It should be mentioned that it is combined with walking share due to the low percentage of cycling among the students surveyed (less than 5%). Students who lived outside of the municipal limit boundaries were omitted from the survey. In addition, questionnaires with missing data, such as ambiguous or no reports of the student’s travel pattern, were removed from the study, and 1260 valid questionnaires were chosen for analysis.

### 3.3. Measurement of BE Indicators and Analysis

This study’s spatial analysis was carried out using a geographic information system (GIS) and a Depth Map. First, the addresses of the participants were coded and precisely populated (at the street and alley levels) in GIS. Then, buffers of 400, 800, and 1600 m surrounding each student’s residence were examined. According to some research studies, a distance of around 800 m is regarded as the threshold for walking [32,36], while others claim that the average daily walk of children to school is about 15 min, or equal to 1600 m [31]. The network distance from the student’s house to the school, Pedshed, land use mix, residential density, average NDVI, and three indicators of connectivity, integration, and choice were then calculated separately from the method of space syntax in all three radii around the student’s house.

The NDVI score [45] was employed in this study to assess the impact of urban green space on active travel. Based on the difference between two bands: plant chlorophyll pigment absorption in the red band (Red) and high reflectance in the near-infrared band (Red), NDVI was used to estimate the gross urban greenness of an entire region using a multispectral picture collection (NIR). The NDVI equation is represented as the following [39]. NDVI values vary from −1.0 to 1.0, with higher values indicating more vegetation cover. To assess total urban greenness, the average value of NDVI in a buffer zone for houses and schools was employed [39].
(1)NDVI=(NIR−Red)/(NIR+Red)

The indicators of space syntax are used to assess the design of urban roadways. The Depth Map was used to analyze the street network design of Shiraz. The spatial structure of the street network was examined using three criteria in this study: Connectivity, Betweenness Centrality (Global Choice), and Closeness Centrality (Global Integration). The number of spaces (streets) that are connected to each segment (space/street) [46,52] and the longer the walk, the greater the value [52]. The sum of the shortest routes between the origin and each destination is used to calculate Centrality. It describes how the space is connected to all other places within a specific radius [5,52]. Betweenness Centrality is a quantity that defines the significance of a node (section/street) in the network. The larger the node’s Centrality, the bigger the number of shortest routes going through it (section/street) [52]. Figure 3 depicts the buffer zone surrounding a home in three desirable radii as an example of connectivity and land use.

One reason for the separation of girls’ and boys’ schools, as well as Iran’s cultural peculiarities, which have prompted parents to safeguard their daughters more successfully than boys, is the Iranian notion of independent transport [15,16]. A binary logistic regression model was used to study the link between three types of explanatory variables: individual characteristics, contingent economic features, the BE, and the patterns of a student traveling to school (Table 3 and Table 4). Collinearity between variables causes inconsistent estimates of each index’s coefficients. According to several research studies, a common cut-off criterion of 5 or less than 10 for the Variance Inflation Factor (VIF), which analyses collinearity, is typically appropriate [56]. The VIF for the variables in the current study is less than 5, indicating that there is no indication of collinearity between the variables. The level of confidence is assumed to be 95 percent. 

## 4. Results and Discussions

### 4.1. Results of Statistical Analysis 

The comparison of walking distances to school for the two genders supports the hypothesis that the AST decision-making process is likely to be gendered (Figure 4). By increasing the walking distance, both genders’ tendency to walk is reduced. While there is no apparent difference between boys and girls for distances under 200 metres, girls find it difficult to walk more than 1400 metres. The 2000 m barrier for boys demonstrates the distinct characteristics of boys, namely their greater physical strength and a better sense of security.

For a more robust comparison, the *Scheffe Test* is used. Boys’ travel behavior between two groups (less than 400 m and over 400 m) is significantly different, meaning distance directly influences their choice of walking (F = 6.477; *p* < 0.002). By increasing the distance above 400 m, boys’ likelihood of walking falls. In other words, boys who live 400 m or more from school are less likely to walk. This test demonstrates that when the walking distance exceeds 800 m, the effect of distance on walking has no meaningful distance for boys.

Three female groupings are similar. Girls under 400 m vary from those above 400 m in travel behavior, although the difference is not statistically significant (F = 2.334, *p* < 0.098) at 95% confidence level. The Scheffe testing shows that there is no statistically significant difference between the three groups of girls.

For determining the most important factors affecting the walking choice of boys and girls to school, Binary Logit models based on three distance thresholds: <400 m, 400–800 m, and >800 m were developed (Table 5).

These data demonstrated that regardless of gender, the urge to walk increases with age. On average, boys are more inclined to walk than girls; the distance has less of an effect on boys’ propensity to walk, demonstrating less elasticity with distance. Our findings revealed a link between the number of student children (7 to 18 years old) in the family and the propensity of the children to walk. In fact, the likelihood of a child walking to school increases as the number of student-aged children in a family grows. This is consistent with the literature [57], as children with two or three siblings were more likely to walk than children with one sibling [26]. By contrast, a previous research study in Iran found no correlation between the number of children in a family and the likelihood of children walking [15].

The model suggests that an increasing number of cars owned by a family lowers the likelihood of walking to school, especially for longer distances to school. This is consistent with the literature, confirming that car ownership had a detrimental impact on children’s active travel to school [28].

Another factor influencing active travel is the number of members in the household who have a driver’s license. The model revealed that increasing the number of persons in the family with a driver’s license reduced the likelihood of walking. The impact of a driving license on walking also increased by increasing the walking distance. This finding is comparable with those of a prior study, which showed that children whose parents did not have a driving license were more likely to walk to school [18].

Although several research studies have found that parents’ travel patterns throughout the day impact their children’s AST [24], our study found that only mothers’ travel habits had a significant association with children’s school travel. 

The walking distance over 800 m was impacted by family income in our study, showing that the walking choice for distances shorter than 800 m is independent of income level. Many research studies [17] have found that higher household income affects the desire to walk.

The results suggest that increasing the distance from school decreases walking and that this effect becomes more evident as the distance increases. These findings are comparable with the findings of other research studies, which show that increasing the distance lowers the likelihood of walking [23]. A study also showed that the travel of children who reside less than 1600 m apart is highly connected to distance [58].

The Connectivity index of the majority of streets in all three radii encourages students to walk. The number of roadway connections increases students’ active travel at all three distance thresholds, although its effect is greatest at shorter distances [59].

As similar results can be found in the literature [44], our study revealed that the effect of street network design weakens beyond a radius of 800 metres, which may be indicative of the additional effort required to walk long distances as a result of the beginning of the effect of distance on the positive impact of street Connectivity [28]. 

In addition, these data suggested that boosting the Choice index within 400 and 800 metres of student residences decreased the amount of walking. Due to the fact that the choice demonstrates the flow in the space, one possible reason for this is that the routes play a large role in shortening the spaces, which, in addition to increasing walking, will increase the amount of through traffic, hence decreasing traffic safety. Interestingly, the effects of Choice on students’ walking are greater for the 400–800 m distance buffer than for the 0–400 m distance buffer.

There was no significant correlation discovered in this study between the Pedshed index, land use mix, NDVI (vegetation), residential density, and Space Syntax measure of Integration. by contrast, two measures of Connectivity and Choice showed significant association with the likelihood of walking although with different patterns for three thresholds of distance [15].

### 4.2. Policy Implications

The study’s findings provide vital information for urban planners and legislators to successfully build areas that encourage AST in children. Because many elements (such as age) are outside policymakers’ control, the suggested regulations’ primary focus is on the qualities of the BE and travel. Our findings indicate that if policymakers want to raise the rate of walking in children’s school travel, gender disparities should be considered, particularly the influence of BE characteristics and street network design on the active travel of children. For example, in recent years, the intensity of demographic shifts and the fall in the number of school-age children in Iran has resulted in the closure or closure of various schools. As a result, the Ministry of Education has implemented a policy of school integration [15]. As demonstrated, the BE, particularly the spatial syntax features, had a substantial influence on the active travel of students. 

According to the findings, it is proposed that school distribution and location should be in areas with higher connectivity (especially with a walking network) and safer from motorized transport. It is feasible to increase the use of active transportation by improving the permeability of the street network [60]. Moreover, the high impact rate of family car ownership variables suggests that transportation demand management push policies would increase the proportion of students in elementary school who walk to school. Moreover, the evidence that students’ walking is highly correlated with the number of cars, driving license holders, and mother’s travel patterns can be attributed to parents’ convenience of using the car and the parents’ safety/security concerns on the way to school. The fact that students chose to walk after a certain distance threshold (particularly girls) indicates that distance does not necessarily influence the decision to drive since many parents choose to drive even for small distances [61].

### 4.3. Study Limitations and Future Works

Although the outcomes of this research study provide new insights into the disparities between individual, interpersonal, and BE characteristics associated with walking to school, it is not without its limits. This research study lacks access to information on the child’s walking route from home to school and the adjacent streets, including building quality, sidewalk width, the presence of urban furniture, and the general design characteristics of the streets and urban areas. In addition, data pertaining to the volume, composition, and speed of motorized traffic would need to be added and analyzed in order to investigate the different effects of street network elements on the walking behavior of students. The effect of cultural and societal conventions, preferences, and habits on the child’s walking is also overlooked. Moreover, parental perspectives and attitudes may impact the choice to encourage children to walk, which needs to be considered in future research [62]. Further, heat-resistant walking pathways are often used for outdoor activities such as commuting to work or school [63]. As it is extensively addressed and assessed in environmental planning literature [64,65,66], future study might focus on establishing an appropriate heat stress threshold for vulnerable populations, such as primary students.

## 5. Conclusions

Physical activity, including walking, is essential for enhancing personal and societal health and advancing sustainable development objectives. Our study focuses on gender disparities in the pattern of active travel (walking) of children to school in the context of Iran as a developing nation, a topic that has received less attention in previous studies. This research study sought to comprehend the pattern of AST in Iran, where schools are divided by gender and a significant part of educational travel is conducted by private car. Due to cultural and religious restrictions, Iranian households treat their girls more conservatively than their boys, hence reducing their freedom and independence in society. Consequently, walking to school may vary by gender, especially in terms of physical qualities such as distance [15]. 

The binary logit model was used to produce three separate models for both genders, each with a distance threshold of three different values. These data demonstrated that regardless of gender, the urge to walk increases with age. Although on average, males are more inclined to walk than girls, the distance has less of an effect on boys’ propensity to walk, demonstrating less elasticity with distance. Within 400 metres, both genders exhibit comparable patterns and are more likely to walk. In the distance range of 400 to 1400 metres, boys are more likely to walk than girls, although the likelihood of walking decreases proportionally with increasing distance for both genders. For distances greater than 1400 metres, girls exhibit no tendency to walk, but boys continue to walk despite a decreasing trend.

The number of school-aged children in a family increases the chance of walking, according to this research study. In addition, these data demonstrated an indirect relationship between family income and the likelihood of children walking to school. However, an increase in income had no discernable effect on walking for distances less than 800 m, but for distances more than 800 m, an increase in income lowered the chance of walking. The number of cars and licensed drivers in a family reduces the likelihood of walking. By extending the distance, it seems that the effects of these two variables are intensifying.

Our studies demonstrated that street network architecture variables, such as Connectivity and Choice, had a significant impact on children’s walking. Intuitively, greater engagement, meeting places with peers, and enhanced accessibility are projected to promote children’s walking propensity [33,67]. For all three distance criteria, the larger the Connectivity, the greater the likelihood that more people will be in the area, which may enhance control monitoring and simplifies entrance, leading to an increase in children’s active travel. Moreover, our data demonstrated that an increase in betweenness centrality (Choice) within 800 m of a house decreases children’s walking owing to concerns about through-traffic safety [22].

Despite similarities in the variables that impact students’ walking, our data indicate that the effect of a number of factors on the walking of primary school boys and girls during their commute to school is distinct. This idea would aid policymakers in considering the gender gap in their direct planning and policies, notably in the BE components. Although this study identified some differences in the characteristics that influence walking to school, the promotion of walking to school among boys and girls is still influenced by a number of multidimentional factors, which is fascinating and calls for more investigation.

## Figures and Tables

**Figure 1 ijerph-20-01131-f001:**
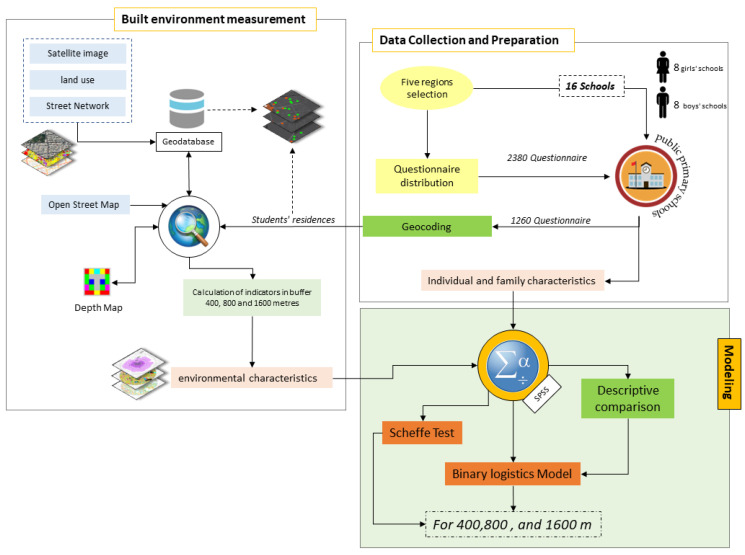
The research process.

**Figure 2 ijerph-20-01131-f002:**
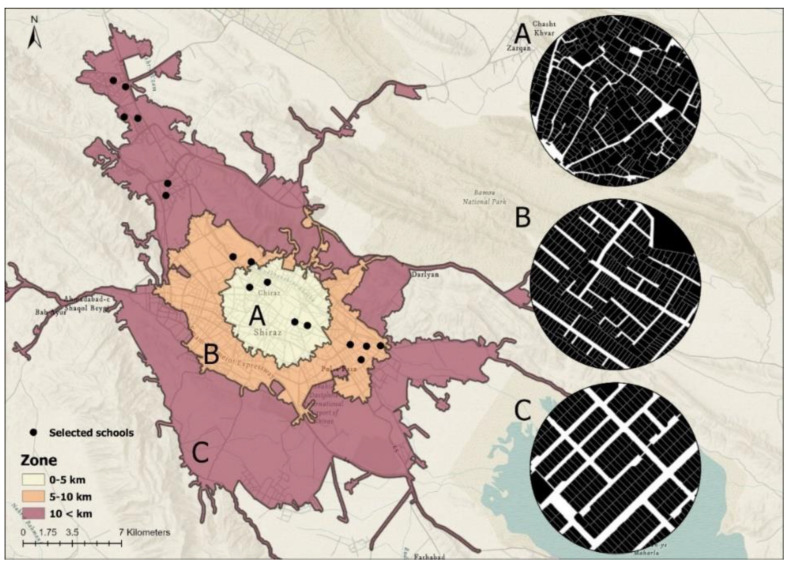
Location of sampled schools. (**A**) Inner zone, (**B**) middle zone, and (**C**) outer zone.

**Figure 3 ijerph-20-01131-f003:**
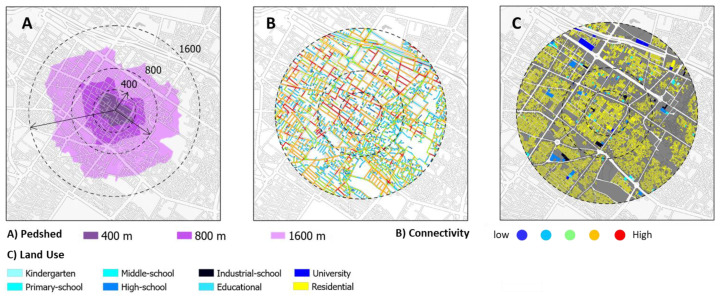
The Pedshed analysis, the Street Connectivity (Space Syntax), and land use around a student’s home.

**Figure 4 ijerph-20-01131-f004:**
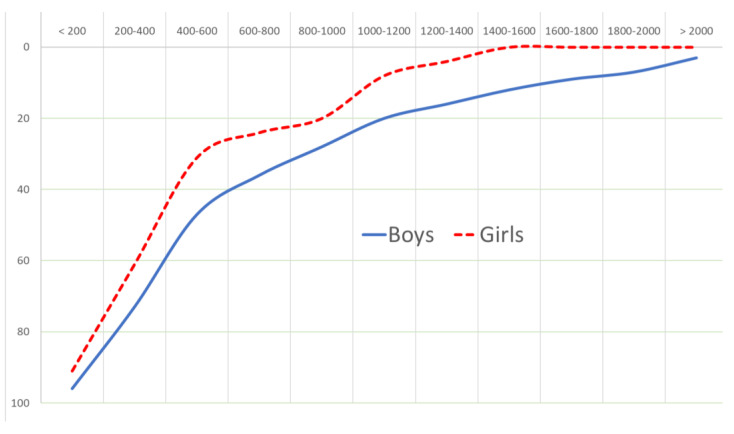
Students’ walking choice decreased by the distance traveled (metres).

**Table 1 ijerph-20-01131-t001:** Summary of selected school’s general characteristics per gender group.

Zone	Municipality District	School	Gender	No. of Sampled Children	Mode
A	2	Dehghani	F	132	11
A	2	Etehadi	M	114	10
A	1	Mostajabi	M	31	10
A	1	Parand	F	63	10
B	1	Aboghadare	F	87	7
B	3	Afife	F	97	7
B	1	Parva	M	66	10
B	3	Askar	M	90	12
B	3	Hesabi	F	51	7
B	3	Imani	M	55	10
C	10	Karimi	F	77	7
C	6	Mosleh Mood	M	68	8
C	6	Mosleh Mood	F	85	7
C	6	Moslem	M	69	10
C	10	Fajr	M	83	10
C	6	Azadi	F	92	10

**Table 2 ijerph-20-01131-t002:** Description of variables used in the study.

Variable	Description	Coding Scale
Active School Travel	The mode of school commuting by the student	1 = Active (walking/cycling), 0 = others,
Age	The age of the student	Continuous variable
Gender (dummy)	The gender of the student	1 = Girl, 0 = others
Father job status (dummy)	The student’s father’s job status is considered Flexible if he is not employed full-time (40 h per week).	1 = Flexible, 0 = others
Mother job status (dummy)	The student’s mother’s job status is considered Flexible if she is not employed full-time (40 h per week).	1 = Flexible, 0 = others
Income	The average family income per month (the value is converted to American dollars)	1 = Less than $595, 2 = $595 to $1190, 3 = More than $1190
Number of children in family	The number of school-aged children (ages 6 to 18) in the family	1 = one child, 2 = two children, 3 = three children or more
Number of cars in family	The number of cars owned by or available to the family	0 = zero car, 1 = one car, 2 = two cars, 3 = three cars or more
Number of driving license holders in family	The number of persons have a valid driving license in the family	0 = no one has license, 1 = on person has license, 2 = two persons have license, 3 = three persons or more have license
Mother traveling mode to work (dummy)	The usual mode of commutting to work by the student’s mother	1 = Active (walking/cycling), 0 = Others
Father traveling mode to work (dummy)	The usual mode of commutting to work by the student’s father	1 = Active (walking/cycling), 0 = Others
Land use mix entropy	The mixture degree of different urban land use classes	Continuous variable
Residential density	The number of dwelling units per gross hectare of residential land area	Continuous variable
NDVI	The index of urban greenness	Continuous variable
Connectivity	The number of (neighbouring) lines that directly intersect a particular axial line	Continuous variable
Integration (closeness centrality)	The index of the relations between each space and all the others in the layout	Continuous variable
Choice (betweenness centrality)	The measure of betweenness of a unit space, that is, the possibility of it being placed at the shortest path connecting spaces in that layout	Continuous variable

**Table 3 ijerph-20-01131-t003:** Walking behavior differences among boys based on distance based on the Scheffe Test.

Distance Group	Distance Group	Mean Difference	Sig.
<400 m	400–800 m	0.230 ***	0.004
	>800 m	0.127 **	0.025
400–800 m	<400 m	−0.230 ***	0.004
	>800 m	−0.103	0.256
>800 m	<400 m	−0.127 **	0.025
	400–800 m	0.103	0.256

Note: *** and ** mean significance at 1 percent and 5 percent levels, respectively.

**Table 4 ijerph-20-01131-t004:** Walking behavior differences among girls based on distance based on the Scheffe Test.

Distance Group	Distance Group	Mean Difference	Sig.
<400 m	400–800 m	−0.090	0.174
	>800 m	−0.008	0.986
400–800 m	<400 m	0.090	0.174
	>800 m	0.082	0.171
>800 m	<400 m	0.008	0.986
	400–800 m	−0.082	0.171

**Table 5 ijerph-20-01131-t005:** Logistic regression estimation among boys and girls.

Variable	400 m				400–800 m				>800 m			
	B	Wald	Sig.	EXP(Or)	B	Wald	Sig.	EXP(Or)	B	Wald	Sig.	EXP(Or)
Age	0.129 ***	243.687	0.004	1.138	0.147 ***	237.102	0.001	1.159	0.129 ***	243.743	0.004	1.137
Gender (dummy)	−0.426 ***	8.270	0.006	0.653								
Number of children in family (7–18 years old)	0.416 ***	7.576	0.001	1.516	0.402 ***	3.551	0.001	1.495	0.419 ***	8.127	0.001	1.521
Number of cars in family	−0.803 ***	11.217	0.000	0.448	−0.754 ***	10.958	0.000	0.471	−0.713 ***	11.654	0.000	0.490
Number of driving license holders in family	−0.322 **	22.155	0.010	0.725	−0.290 **	20.146	0.020	0.748	−0.285 **	17.744	0.025	0.752
Mother traveling pattern (dummy)	0.427 **	6.618	0.040	1.532								
Family income group									−0.369 **	5.043	0.026	0.691
Distance (network) to school	−0.850 ***	20.659	0.000	0.428	−0.843 ***	5.453	0.000	0.430	−0.831 ***	5.228	0.000	0.414
Connectivity < 400 m	0.483 ***	32.374	0.000	1.622								
Choice < 400 m	−0.327 ***	4.211	0.000	0.721								
Connectivity 400–800 m					0.447 ***	38.025	0.000	1.727				
Choice 400–800 m												
Connectivity > 800 m									0.359 ***	17.897	0.000	1.432
Choice > 800 m					−0.180 **	3.139	0.022	0.835				
Overall prediction percentage	80.2				80.6				81.0			
−2 Log likelihood	1073.996				1095.815				1064.783			
Nagelkerke R square	0.502				0.493				0.496			
Omnibus test	Chi-square		Df (sig.)		Chi-square		Df (sig.)		Chi-square		Df (sig.)	
	570.750		9 (0.000)		562.431		7 (0.000)		552.053		8 (0.000)	

Note: *** and ** mean significance at 1 percent and 5 percent levels, respectively.

## Data Availability

The summary of raw datasets is available from the corresponding author.

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
