# Peer review of "How the Built Environment Moderates Gender Gap in Active Commuting to Schools"

_ijerph, 2023, doi:10.3390/ijerph20021131_

Round 1
Reviewer 1 Report
A referee report on “
How the Built Environment Moderates Gender Gap in Active Commuting to Schools”
General Comments
1. The introduction section is well-crafted, serving also as a decent literature overview. However, there are numerous English errors (see specific comments below) that need to be corrected. The use of semicolons should also be reduced to a minimum. I started to refer to the English errors line by line, but they are so numerous that I suggest a native English speaker to edit the text.
2. The contribution of this study is only mentioned at the end of the Introduction section. Why make the reader wait for so long? It should be described in the first two paragraphs. Alternatively, reduce Introduction and add a Literature Review section to keep the exposition more structured.
3. Section 2.3 titled “Analysis” is just one paragraph. The authors should better blend this “section” with the other sections in the text.
4. The exposition of the results lacks clarity. For instance, Figure 3 and Tables 3 to 5 are all lumped together, which makes it very difficult to read. The way statistical significance is presented is also rather unconventional. I would advise reporting standard errors below the estimated coefficients with the asterisks such as e.g. ** representing the level of statistical significance: *** for 1% etc.
Specific Comments
1. The abstract is too long, it could be made more informative by focusing on the most important contribution of the study
2. Lines 44, 45: excessive use of “furthermore”
3. Line 54: “anxiety"-->anxiety relief maybe? Same for obesity and relaxation in the next line.
4. Line 61: researchàresearch studies
5. Line 75: the study indicatedàone study indicated
6. Line 86: according to previous studies
7. Line 112: a increasingàan increasing
8. Line 139: previous studyàprevious studies
9. Line 199: while in Iran it is $0.06
10. Line 148: equivocalàreplace with a simpler word
11. Line 162: a lot of studyàa lot of studies
12. Line 246: the student’s à the students’
13. Line 252: capital T should be made small case
Reviewer 2 Report
This manuscript is investigating the influences of built environmental factors, network design, and sociodemographic factors on active travel to school. Although numerous studies have explored these relationships, this study is trying to assess this issue with a focus on gender differences. The topic of this manuscript is within the scope of the International Journal of Environmental Research and Public health, and it brings up some novelty, but several major flaws are preventing its publication.
1-Although the language is well, there are some typos and some of the paragraphs are not clear. It needs some cleaning and editing. Some of the main issues are as follows:
1) In lines 45 to 46, The reference [2] should be only at the end of the sentence and the first reference [2] can be removed.
2) Line 47: The term “infancy” is not proper for ages 6 to 14 and it is better to be changed it to childhood.
3) Line 66: AST selection is better to be changed to AST choice.
4) Line 68: guys should be changed to boys.
5) Line 83: The meaning of this sentence is not clear: “In this case, the evidence is ambiguous.”
6) Line 87: The term “autonomous movement” is not clear.
7) Line 126: “Some researches” should be changed to “Some research studies”
8) Line 200: The meaning of “Scattering, as well as uneven development and organisation of various purposes around the city, adds another layer of complications” is not clear.
9) Line 223: “Schools were chosen from all three zones in order to serve as a model for the country's current urban surroundings.” The current study area is not sufficient as a representative of a whole country regarding the built environmental and network design factors.
10) Figure 2: The word “laand use” should be changed to “Land Use”
11) Line 334: Table 3 should be changed to Table 4.
2- In some parts of the manuscript, the authors claim that several cultural factors may affect girls’ choices of walking to school, and generally it is because parents' may prevent their daughters from choosing to walk to school. Based on the contents of this manuscript the authors did not provide enough evidence and references related to that claim. For example from lines 171 to 173, the authors mentioned that “Although there are no limits on women's physical activities and social contacts in Iran, Middle Eastern families may be more conservative about their daughters and do not allow their daughters to go to and from school as freely as boys [30,76]”. Or from lines 303 to 306: “As previously noted, one of the reasons for this is the separation of girls' and boys' schools, as well as Iran's cultural peculiarities, which have prompted parents to safeguard their daughters more successfully than boys in the Iranian notion of autonomous transportation [30, 76]”. And also lines 337 to 339: “This might be because parents are more cautious about females, regardless of age, utilising active modes of transportation to school”. From lines 352 to 355 “Nonetheless, the family's affluence has had little influence on the travel of male students. This may be an emphasis on why parents are more rigors about their daughters' attendance independently or using active transportation such as walking to school, and it increases the likelihood of females walking to school to drop”. In all these parts, the authors expand their conclusion about this issue to all middle eastern countries while they are generally citing only two studies that are only related to Iranian cities. Furthermore, these cited studies are not published in social and psychological, or cultural journals and are not good references for this conclusion. Since these are just the authors' claims and there is no visible evidence to prove them based on the manuscript outcomes, these parts are better to be omitted or revised. In addition, this issue is not well merged with the journal’s goals.
3- In lines 180 to 181, the authors mentioned that “So yet, no study has independently as-180 sessed the indicators of space syntax on the AST for girls and boys”. This is again the author's claim and for proving that they need to provide a systematic literature review. So, it is recommended to be revised.
4- The methodology section is not clear. Section 2 (Materials and Methods) is supposed to clarify the methodology and modeling process and shows how the authors try to integrate the results from GIS with the binary logit models. There is a need to add a section that explains the modeling.
5- In Section 2.2 Sampling, the authors need to explain how they choose their sample size. Is it based on a scientific method? Why 1260 responders are sufficient for this study?
6- There is an inconsistency between the sample size. 1) in line 230 the authors mention that the data is gathered from students in the first to sixth grades (6 to 14 years old), however, based on the table (1) the mean age of students is between 9 to 11. I suggest that line 230 be revised based on table 1. 2) Although the collected data is almost uniform by gender (684 girls and 666 boys based on table 1), this data is not uniform based on age (age 9: 242 boys and 529 girls, age 10: 335 boys and 0 girls, and age 11: 69 boys and 155 girls). This inconsistency would lead to a misconclusion. In their sample size, for example, the authors determined that girls decided to AST less than boys. In Tables 3 and 4, however, age has a positive effect on AST. As a result, younger children are less likely to choose to walk to school, and the majority of data collected from female students is for the age of 9. The authors must clarify this fundamental problem and alter their data based on a uniform sample size while taking the age factor into account to conduct a better analysis.
7-Based on Figure 3 (Students’ modal choice of girls and boys by distance traveled), the survey's sample size is divided based on 400 meters, however, in tables 3, 4, and 5, analysis results are presented based on distances 400, 800 and 1600 meters (based on the GIS buffers): 1) The proportion of the sample size population is not clear for each distance category shown in Figure 3. How many sample sizes are gathered for distance ranges of 400, 400-800, 800-1200, and greater than 1200 meters? 2) The percentage of the sample size population who chose active travel is dropping significantly from a distance greater than 800. This drop may result in inaccurate analysis results. It seems that the authors only used the sample size population included in the distance categories of 800 to 1600 meters, and ignored the responses in the distance range of 1600 meters to more than 2000 meters (Based on Tables 3, 4, and 5). This will make the analyzed sample size to be less than 1260, which is another contradiction that the authors need to clarify. For better analytical findings, I recommend that the authors use all data from 800 to more than 2000 meters.
8-The section 3.4 should be changed to study limitations and future works.
